# Revisiting cysteine protease function in *Trypanosoma cruzi*: implications for parasite egress and differentiation

Sara De Grandis,[1,2] Anne Niggli,[1] Delia Bogenstätter,[1] Gaelle Lentini[1]

**ABSTRACT**    Chagas disease is a major global health concern affecting millions of people worldwide, with limited therapeutic options in its chronic phase and no prophylactic vaccine. The causative agent, *Trypanosoma cruzi*, is a unicellular eukaryotic parasite whose life cycle alternates between insect vectors and a wide range of mammalian hosts. In mammalian cells, parasite proliferation depends on iterative cycles of host cell invasion, intracellular multiplication, differentiation, and host cell rupture, releasing hundreds of infective parasites. The mechanisms governing the critical transition from replicative amastigotes to infective trypomastigotes (trypomastigogenesis) and subsequent egress remain poorly understood, largely due to the lack of robust analytical tools. Here, we combined real-time cellular impedance monitoring, stage-specific fluorescent parasites, ultrastructural expansion microscopy, and automated high-content imaging to dissect the late steps of the lytic cycle. We provide quantitative evidence that trypomastigogenesis is temporally coordinated with egress, ensuring the release of fully mature, infective trypomastigotes. Furthermore, we re-evaluate the effect of the cysteine protease inhibitor Z-Phe-Ala fluoromethyl ketone (Z-FA-FMK) at late stages of infection. Our results quantitatively support previous observations that Z-FA-FMK impairs trypomastigogenesis, causing an accumulation of amastigotes and blocking progression to mature trypomastigotes. This arrest delays egress and leads to the release of immature forms, highlighting the essential role of cysteine proteases in parasite differentiation. Together, our work establishes a quantitative framework for dissecting the tightly regulated, multi-step process of lytic cycle termination in *T. cruzi* and offers a versatile platform for phenotypic screening and drug discovery.

**IMPORTANCE** Chagas disease, caused by *Trypanosoma cruzi*, affects millions worldwide and remains a major global health burden, causing chronic cardiac, digestive, and neurological complications. The disease, disproportionately impacting vulnerable populations, lacks effective treatments for the chronic phase of the disease or a vaccine for its prevention. Parasite replication and host cell exit are tightly linked, but the mechanisms driving the transition from intracellular replicative parasites to infective forms and their subsequent release upon host cell lysis are poorly understood. Using real-time monitoring, fluorescent parasites, and high-resolution imaging, we provide quantitative evidence that cysteine proteases are critical for parasite maturation and that their inhibition uncouples differentiation from egress, leading to the release of immature parasites that are generally considered less infective. These findings reveal fundamental principles of parasite biology, provide a platform for drug discovery, and highlight new avenues to target Chagas disease at a critical stage of infection.

**KEYWORDS**    egress, host-pathogen interactions, *Trypanosoma cruzi*, differentiation, Chagas disease, neglected tropical diseases, protease inhibitors

Address correspondence to Gaelle Lentini, gaelle.lentini@unibe.ch.

The authors declare no conflict of interest.

See the funding table on p. 16.

*T*rypanosoma cruzi, the causative agent of Chagas disease, is a protozoan parasite with a complex life cycle that alternates between insect vectors and mammalian hosts. Within the mammalian host, *T. cruzi* exhibits a remarkably broad cellular tropism, capable of infecting virtually all nucleated cells, across a wide range of warm-blooded animals. This adaptability contributes to a vast sylvatic reservoir that sustains transmission across diverse ecological niches. Globally, Chagas disease affects an estimated 10 million people, with chronic infections leading to severe cardiac and gastrointestinal complications. Despite its widespread impact, the disease remains neglected and continues to pose a major public health challenge, with limited therapeutic options for the chronic phase of the disease and no effective vaccine for its prevention (1, 2).

For this obligate intracellular parasite, the accomplishment of its lytic cycle is essential for parasite dissemination and disease progression. *T. cruzi* invasion of non-phagocytic cells relies on lysosome-dependent and -independent pathways, accompanied by rearrangement of host cell actin (3–5). Depending on the strains considered, the parasite engages different surface signaling molecules to trigger F-actin disassembly or recruitment that ultimately leads to the internalization of the pathogen in a parasitophorous vacuole (PV) (4, 6, 7). Within the first 18 h post-infection, the parasite mediates its escape from the PV and establishes a cytoplasmic residence. During this step, the parasite undergoes a series of morphological and functional transformations in a process termed amastigogenesis, which describes the transition from trypomastigotes to amastigotes, the replicative form of the parasite. This transformation is characterized by a decrease in size of the parasite body and flagellum, the displacement of the kinetoplast from the posterior to the anterior side of the cell with respect to the parasite nucleus, and by extensive remodeling of the parasite transcriptome and proteome (8–11). The acidification of the PV, through the fusion with host lysosomes, seems to be crucial for both amastigogenesis and PV escape (12–14). Following a period of intracellular multiplication in the host cytosol, amastigotes undergo trypomastigogenesis, differentiating into motile trypomastigotes primed for dissemination (8). Once again, this differentiation is characterized by important morphological changes including the elongation of the flagellum and posterior cell body, the acquisition of an undulating membrane, and the translocation of the kinetoplast to a basal position relative to the nucleus. During this differentiation, intermediate forms sharing some structural and antigenic properties with epimastigotes have been observed (8, 15). Preliminary studies suggest roles for parasite IP3R-mediated $Ca^{2+}$ signaling and host L-proline in this process (16, 17). Moreover, proteasome inhibitors have been shown to affect *T. cruzi* stage differentiation, thus highlighting the crucial role of proteasome-mediated proteolysis during parasite remodeling (18–20).

Egress, the exit of *T. cruzi* from its host cells, appears to be a multifactorial process, likely involving parasite motility, parasite-derived factors, host cell signaling pathways, and extensive remodeling of the host cytoskeleton, which destabilizes shortly before cell rupture (21–24). The culmination of these events leads to the breakdown of the host cell membrane and the release of infective trypomastigotes into surrounding tissues. Prior studies based on pharmacological inhibition suggest that cysteine proteases, including cruzipain, facilitate parasite escape from host cells, as treatment of infected cultures with reversible or irreversible cysteine protease inhibitors reduces the number of trypomastigotes released into the culture supernatant (23, 25). These inhibitors have also been reported to interfere with stage transitions, from trypomastigote to amastigote and from amastigote to trypomastigote, depending on the timing of treatment, while exerting only moderate effects on parasite multiplication (25, 26). Consequently, it remains unclear whether the reduced parasite egress observed upon cysteine protease inhibition reflects a direct role of these enzymes in host cell rupture or, alternatively, an indirect consequence of impaired intracellular parasite maturation and differentiation.

Despite the importance of egress in *T. cruzi* pathogenesis, its study has been limited, partly due to the absence of phenotypic tests assessing egress efficiency in a rigorous and quantitative manner and the difficulty of uncoupling trypomastigogenesis from

parasite release. The quantification of egress has been mostly assessed by manual quantification of released parasites, a labor-intensive, non-scalable approach prone to experimenter bias (21, 23, 24, 27). Similarly, revealing defects in trypomastigogenesis relies on Giemsa staining, which provides limited resolution and hampers comprehensive morphological analyses (26). As a result, our understanding of the cellular and molecular mechanisms underlying parasite egress remains fragmented.

In this study, we investigate the temporal dynamics of parasite differentiation and egress, the role of cysteine proteases, and the morphological transitions associated with trypomastigogenesis. By integrating ultrastructural analyses and pharmacological inhibition with multi-parametric, quantitative phenotypic measurements, we establish a solid, time-efficient analytical pipeline to study egress and trypomastigogenesis. This approach provides quantitative, time-resolved measurements that extend and refine previous seminal observations, thereby bridging descriptive and quantitative analyses of parasite differentiation and release (8, 22, 23, 26). Using this pipeline, we revisit the role of Z-Phe-Ala fluoromethyl ketone (Z-FA-FMK), an irreversible cysteine protease inhibitor, and show that it acts as an early-stage trypomastigogenesis inhibitor. Altogether, our work provides a comprehensive framework for elucidating key mechanisms underlying parasite differentiation and egress and for identifying novel therapeutic targets aimed at halting parasite dissemination.

## RESULTS

### *T. cruzi* egress is a temporally controlled event synchronized with parasite development

The exit of *T. cruzi* from its host cell marks the end of the lytic cycle. This step is characterized by the rupture of the infected host cell and the release of parasites into the extracellular space. We employed measurements of cellular impedance, expressed as cell index (a unitless, normalized value derived from impedance), to monitor cellular events, in real-time, of human foreskin fibroblast (HFF) culture infected by *T. cruzi* (Fig. 1A). During the initial phase post-invasion (0- to 0.10-day post-infection [dpi]; corresponding to ~2.5 h post-invasion [hpi]), the cell index of infected cultures transiently decreased from 100% to 71.88 % (±12.56, $P = 0.0323$) compared to uninfected cells. This sharp decrease in the electrical impedance associated with the infection is likely due to the parasite's manipulation of host cytoskeletal structures during the invasion process (6, 7). Between 0.58 and 3.45 dpi, the cell index of infected culture appeared slightly higher than the control; however, not significantly different. Around 4 dpi, the cell index of the infected culture started to be significantly lower than the control and continuously decreased over time to reach a plateau at 6 dpi (~27% compared to control) (Fig. 1A and B). The number of parasites released in the extracellular space upon host cell rupture, over the course of the infection, was assessed by manual counting. The peak of egress is observed at 6 dpi corresponding to the maximal host cell lysis measured by cellular impedance (Fig. 1C). However, we noticed a temporal discrepancy between the observed decrease in cell index (starting at 4 dpi) and the release of parasite from the infected host cells (Fig. 1A and C). As previously reported, egress is preceded by the differentiation of the amastigotes into trypomastigotes (trypomastigogenesis), during which different forms are observed (8). To resolve parasite differentiation kinetics in a quantitative manner, we employed Ultra-Expansion Microscopy (U-ExM) to visualize individual intracellular parasites within infected cells from 4 to 6 dpi. Infected cells were classified into three categories based on the predominant form adopted by the intracellular parasites within each host cell: amastigotes (round, short flagellum, and anterior kDNA), intermediate forms (ovoid, longer flagellum, kDNA anterior, or migrating) and trypomastigotes (slender, wavy flagellum attached to the cell body, and kDNA posterior) (Fig. 1D). At 4 dpi, nearly all infected cells contained amastigotes (98.70 ± 1.44%). However, at 5 dpi, this proportion dropped to 52.80% (±5.38) (Fig. 1E). At this stage, a large proportion of infected cells harbored parasites in an intermediate form (38.30 ± 6.14%), while few contained parasites that had already completed trypomastigogenesis. The number of

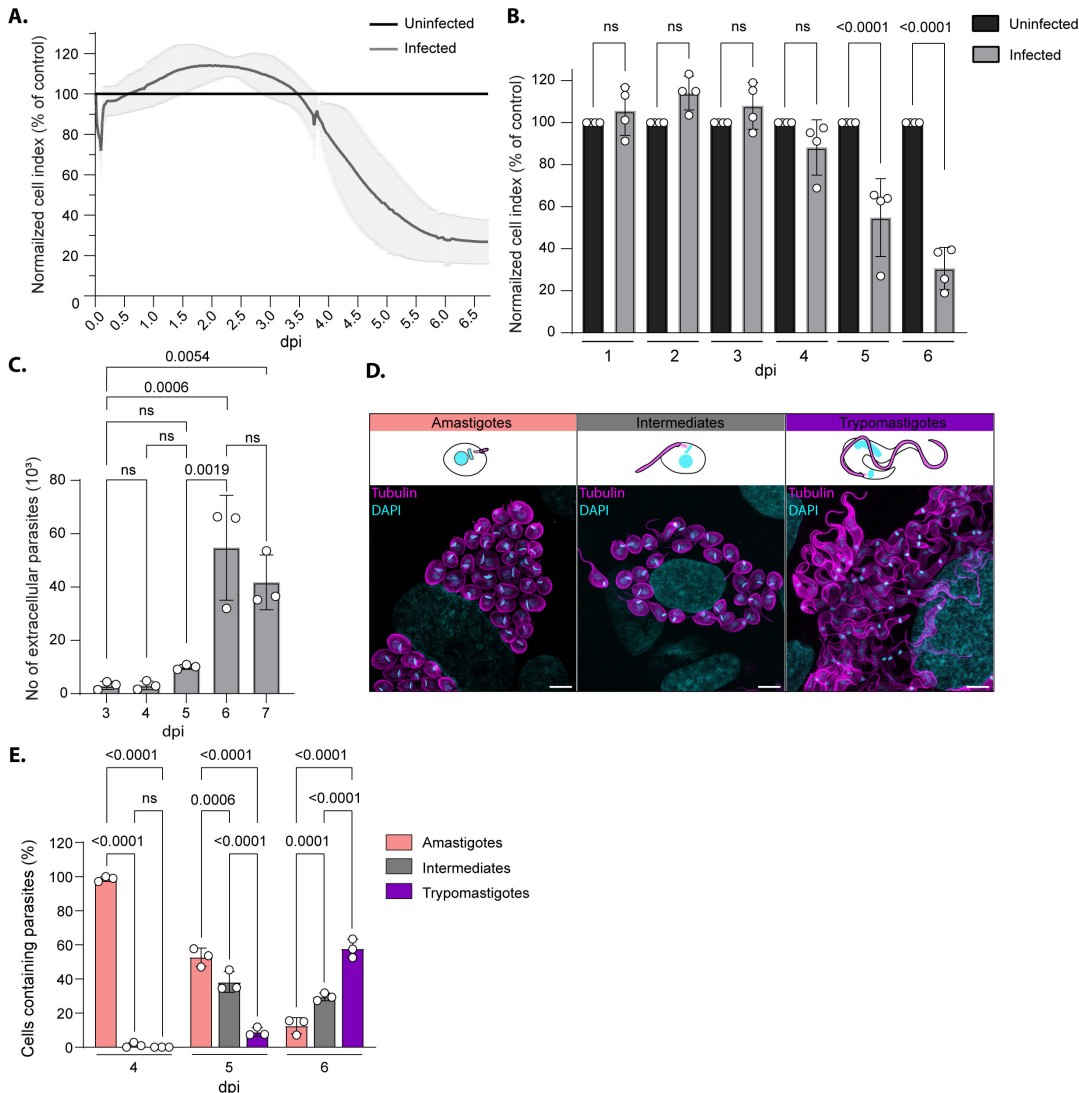

**FIG 1** *T. cruzi* egress is coordinated with trypomastigogenesis. (A) Kinetic measurement of host cell index over the course of *T. cruzi* infection normalized to uninfected culture (mean ± SD, *n* = 4). (B) Graph representing the cell index normalized to uninfected host cell monolayer day by day (mean ± SD, *n* = 4). Statistical significance was assessed by a one-way ANOVA with Tukey's multiple comparison. (C) Graph representing the number of extracellular parasites as assessed by manual quantification at 3, 4, 5, 6, and 7 dpi (mean ± SD, *n* = 3). Statistical analysis was done using ordinary one-way ANOVA followed by Tukey's multiple comparisons test. (D) Ultra-expansion microscopy images of infected cells and schematic representation of the different *T. cruzi* forms observed during trypomastigogenesis. DNA was stained with 4',6-diamidino-2-phenylindole (DAPI, cyan), and parasites were visualized using anti-tubulin antibodies (magenta). Scale bar = 20 µm. (E) Graph representing the quantification of the different intracellular forms observed at 4, 5, and 6 dpi (mean ± SD, *n* = 3). Statistical significance was assessed by a two-way ANOVA with Tukey's multiple comparison. ns, not significant.

parasites detected in the culture supernatant at 5 dpi did not differ significantly from that observed at 3 or 4 dpi, indicating that parasites remained largely intracellular. By 6 dpi, most infected cells were filled with fully differentiated trypomastigotes (57.88 ± 5.46%) (Fig. 1E). At this point, many parasites had already egressed from the host cells, mostly as trypomastigotes (Fig. 1C; Fig. S1A and B). These results suggest that events other than host cell lysis must account for the initial drop in electrical impedance between 4 and 5 dpi, potentially including trypomastigogenesis. Altogether, these results support the notion that the late phase of the lytic cycle during *T. cruzi* infection is a multi-faceted process timely controlled, initiated by parasite differentiation from 4 to 6 dpi and subsequent egress of differentiated forms.

## Development of an automated high-content imaging platform for studying *T. cruzi* egress

To date, egress efficiency of *T. cruzi* from infected cultures is typically assessed by manually counting extracellular parasites released in the culture supernatant (16, 21). Studies of the molecular mechanisms involved in this process face formidable technical challenges, requiring labor-intensive manual identification and counting, which is subject to experimenter bias. To overcome this, we developed an automated high-content imaging method to assess egress efficiency in *T. cruzi*. We first tested this method on samples previously quantified by manual counting (Fig. 1C). Culture supernatants were collected at different dpi in a 384-well plate, fixed, and stained with Hoechst to visualize parasite DNA. Samples were imaged and the number of parasites in the culture supernatant was quantified using an automated imaging and analyzer system. Briefly, 12 images were acquired per well (covering ~85% of the well area) using a 20× objective and the DAPI channel of the IN Cell Analyzer 2000 (Cytiva). Quantification of Hoechst-positive parasites was performed using the IN Cell Developer software (Cytiva). Because *T. cruzi* contains two spatially separated DNA-containing structures, the nucleus and the kinetoplast, both labeled by Hoechst, automated segmentation occasionally identified these structures as independent objects, leading to overestimation of parasite numbers. To address this, sensitivity and smoothing thresholds were empirically optimized to count closely associated Hoechst-positive objects as a single parasite. Increasing the merging distance reduced over-segmentation of individual parasites but also increased the likelihood of erroneously merging closely adjacent independent parasites, particularly at higher parasite densities. Therefore, a conservative threshold was selected to balance these opposing effects (see Materials and Methods). Potential Hoechst-positive host cell nuclei or debris were excluded using a size filter (>3 and <30 $\mu m^2$, Fig. S2A). Consistent with the results obtained by manual counting, the number of extracellular parasites increased significantly at 6 dpi (Fig. 1C and 2A). In addition, compared to manual counting, the automated method demonstrated enhanced reproducibility and stronger statistical power. We subsequently explored the utility of this approach to discriminate egress-inhibiting compounds. Following the intracellular multiplication phase of the parasite (4 dpi), the culture supernatant was collected and replaced with media containing drugs. Two drugs were tested here. Benznidazole (BZN), which triggers DNA damage, cell cycle arrest, and parasite death, and Z-FA-FMK, a potent irreversible cysteine protease inhibitor, previously reported to inhibit *T. cruzi* egress (23, 28, 29) (Fig. 2B). The toxicity of both drugs on the host cells was assessed by cellular impedance measurement. While Z-FA-FMK treatment (50 µM) did not cause detectable deleterious effects in either infected or uninfected cells, BZN treatment at 25 µM strongly impacted cellular impedance shortly after its addition (96 hpi) (Fig. S2B through D). This cytotoxic effect was only observed on *T. cruzi*-infected cultures but not on uninfected cells, consistent with the enzyme-mediated activation of the prodrug by the parasite nitroreductase (30) (Fig. S2E and F). Titration of BZN on infected cells revealed that a concentration of 3.25 µM did not trigger a premature decline in cellular impedance, unlike the higher concentrations of 12.5 and 25 µM, which caused an early drop (Fig. S2F). Therefore, we used BZN at 3.25 µM in our assays. Culture supernatant at 4 dpi (0 h of drug treatment) and 6 dpi (48 h of drug treatment) was analyzed. At 4 dpi, prior to drug treatment, the number of egressed parasites was comparable in all the cultures as assessed by both methods (Fig. 2C; Fig. S2G). At 6 dpi, the number of extracellular parasites was reduced by 89.60% (±2.24) in Z-FA-FMK-treated cultures and by 62.60% (±4.15) in the BZN-treated cultures (Fig. 2D). Comparable trends were observed using manual counting, albeit with reduced reproducibility (Fig. S2H). Kinetic measurements of cellular impedance showed a moderate effect of Z-FA-FMK treatment on infected cultures (Fig. 2E). During the first 4 days of infection, no difference could be observed between the different cultures prior to treatment (Fig. 2E, black arrow and Fig. 2F). At 6 dpi, the cellular impedance of infected cultures treated with Z-FA-FMK was significantly higher than that of the untreated infected control. However, Z-FA-FMK treatment did

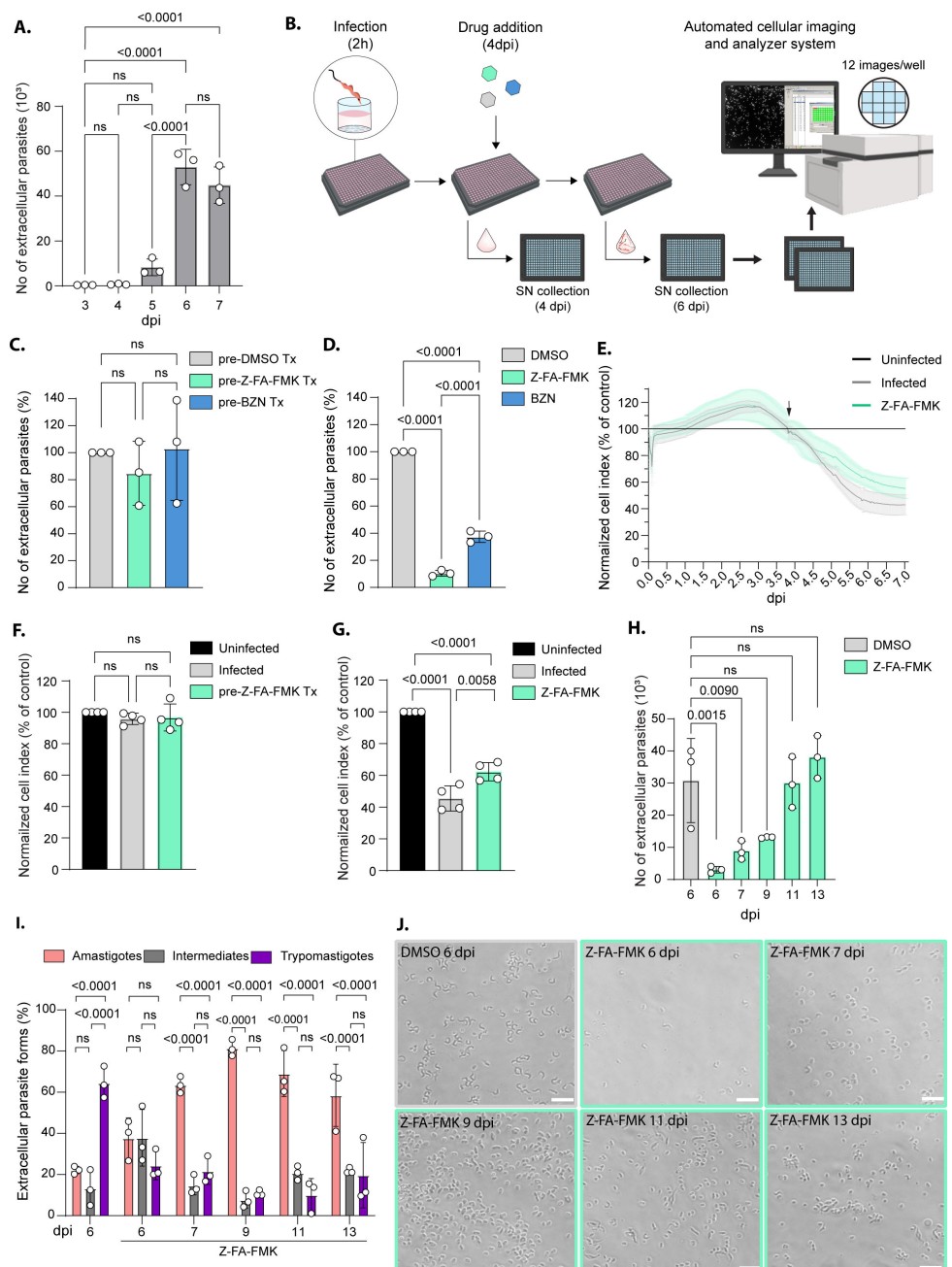

**FIG 2** Validated workflow to measure egress efficiency in *T. cruzi*. (A) Graph showing the quantification of extracellular *T. cruzi* at 3, 4, 5, 6, and 7 dpi using the automated image analysis pipeline. Statistical significance was assessed by a one-way ANOVA with Tukey's multiple comparison (mean ± SD, *n* = 3). (B) Scheme of the semi-automated workflow used to quantify *T. cruzi* egress efficiency. (C) Graph showing the automated quantification of extracellular parasites at 4 dpi in cultures prior to treatment (Tx). Statistical analysis was performed using ordinary one-way ANOVA with Tukey's multiple comparison (mean ± SD, *n* = 3). (D) Graph showing the automated quantification of extracellular parasites at 6 dpi in cultures treated with Z-FA-FMK or BZN, normalized to DMSO control. Statistical analysis was performed using ordinary one-way ANOVA with Tukey's multiple comparison (mean ± SD, *n* = 3). (E) Kinetic measurement of cell index over the course of *T. cruzi* infection in culture treated or not with Z-FA-FMK normalized to uninfected culture (mean ± SD, *n* = 4). (F) Graph representing the cell index of infected culture pre-treatment (Tx) normalized to uninfected culture at 4 dpi (mean ± SD, *n* = 4). Statistical significance was assessed by a one-way ANOVA with Tukey's multiple comparison. (G) Graph representing the cell index of infected culture treated or not with Z-FA-FMK normalized to uninfected culture at 6 dpi (mean ± SD, *n* = 4). Statistical significance was assessed by a one-way ANOVA with Tukey's multiple comparison. (H) Graph showing the automated quantification of extracellular parasites

Fig 2 (Continued)

at 6, 7, 9, 11, and 13 dpi in infected culture treated with Z-FA-FMK compared to control (DMSO, 6 dpi). Statistical analysis was performed using ordinary one-way ANOVA with Tukey's multiple comparison (mean ± SD, $n$ = 3). (I) Graph showing the manual quantification of extracellular *T. cruzi* forms at 6, 7, 9, 11, and 13 dpi in cultures treated with Z-FA-FMK compared to control condition (DMSO, 6 dpi). Statistical analysis was performed using two-way ANOVA followed by Tukey's multiple comparison (mean ± SD, $n$ = 3). (J) Widefield images of the supernatant of infected cultures at 6, 7, 9, 11, and 13 dpi treated with Z-FA-FMK compared to control condition (DMSO, 6 dpi). Scale bar = 20 µm. ns, not significant.

not fully restore impedance to the level of uninfected controls, suggesting that other infection-related events impact cellular impedance (Fig. 2G). Beyond six dpi, the cell index in Z-FA-FMK-treated cultures continued to decline without reaching a plateau (Fig. 2E). Thus, we quantified the number of parasites released into the culture supernatant beyond the typical duration of the lytic cycle. Both manual and automated counts revealed a continued release of parasites beyond 6 dpi in Z-FA-FMK-treated cultures, reaching a level comparable to the control (DMSO, 6 dpi) by 9 dpi (Fig. 2H; Fig. S2I). Notably, examination of the released parasites in the culture supernatant revealed predominantly round forms, likely corresponding to amastigotes (Fig. 2I and J).

## The cysteine protease inhibitor Z-FA-FMK affects trypomastigogenesis

To investigate the impact of Z-FA-FMK treatment on intracellular parasite differentiation, a double-reporter TdTom/ama-mNG strain was generated. In this strain, TdTomato (TdTom) fluorescent protein is fused to the 3′UTR of *gapdh* ensuring constitutive expression in the different life forms of the parasites while fusion of mNeonGreen (mNG) to the 3′UTR of amastin ensured robust expression of mNG, specifically in the amastigote stage (Fig. 3A; Fig. S3A and B) (31). Observation of cells infected with TdTom/ama-mNG parasites showed that most of the intracellular parasites expressed mNG at 4 and 5 dpi. However, parasites started to lose mNG fluorescence at 6 dpi, and some infected cells appeared mostly devoid of mNG fluorescence (Fig. 3B and C, white arrowhead). Quantification showed that, at 6 dpi, around 50% of the infected cells contained predominantly mNG-negative (mNG−) parasites, indicating that most parasites have undergone trypomastigogenesis. In contrast, in Z-FA-FMK-treated cells, 90.38% (±3.38) of the infected cells remained mNG-positive (mNG+), suggesting that the intracellular parasites did not successfully accomplish trypomastigogenesis (Fig. 3D). A defect in parasite multiplication might account for a delay in the transition from amastigotes to trypomastigotes. Quantification of the number of intracellular parasites at 5 dpi (prior to egress) showed only a moderate defect in parasite multiplication under Z-FA-FMK treatment conditions that cannot fully explain the drastic effect observed on the retention of the mNG fluorescence (Fig. 3E). Altogether, these findings are consistent with previous reports that Z-FA-FMK treatment affects trypomastigogenesis, causing delayed egress and preferential release of immature forms. They also highlight the utility of this dual-reporter strain for quantitatively monitoring trypomastigogenesis via molecular markers.

## Immature forms are arrested at the early stage of trypomastigogenesis

We employed U-ExM to investigate in greater detail the impact of Z-FA-FMK treatment on trypomastigogenesis. Infected cultures were either treated with Z-FA-FMK at 4 dpi or left untreated, and parasite forms were quantified at 5 and 6 dpi as previously described (Fig. 1D). At 5 dpi, most infected cells contained either amastigotes or intermediate forms, whereas only a small proportion harbored trypomastigotes in both conditions (Fig. 4A). Consistent with observations from the dual-reporter strain experiment, a marked accumulation of amastigote-containing cells was detected at 6 dpi in Z-FA-FMK-treated cultures (56.16 ± 1.00%), compared to the control (12.58 ± 4.7%). This increase in amastigote forms was accompanied by a reduction of cells containing more mature forms (intermediate and trypomastigotes) (Fig. 4B). To assess potential morphological

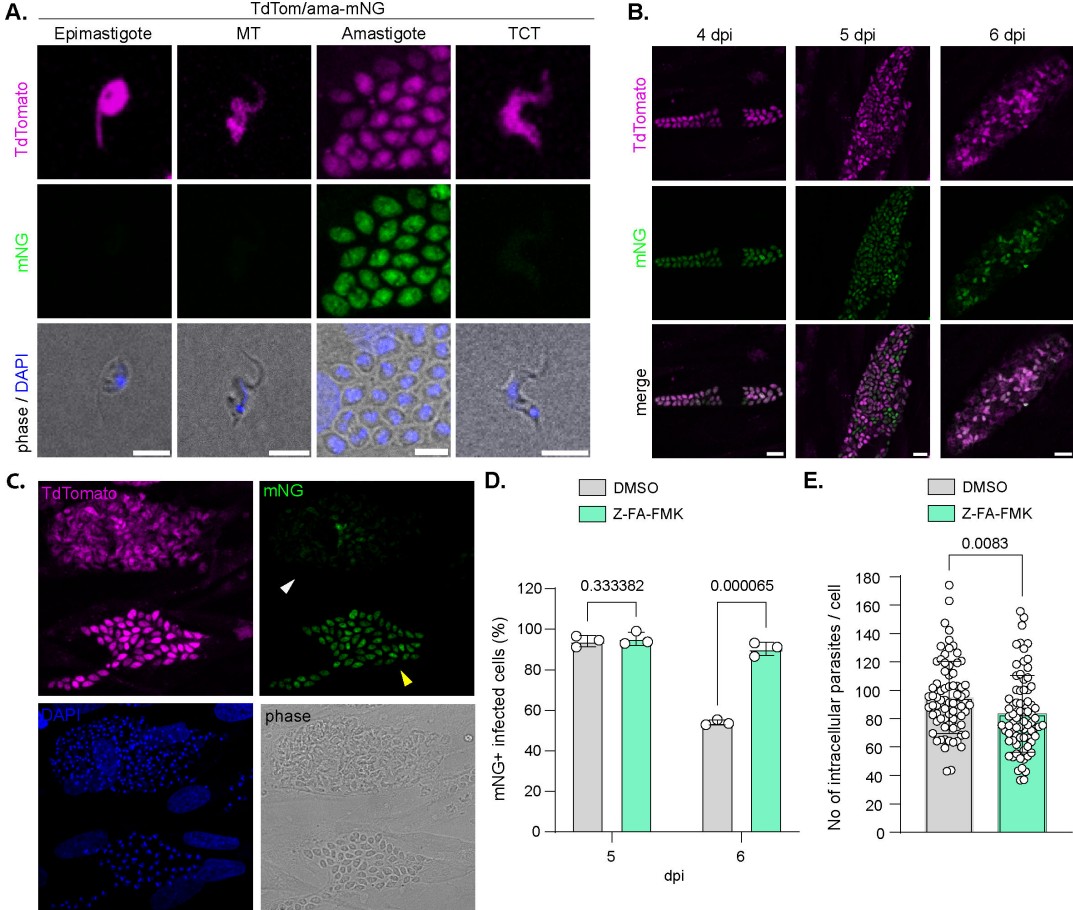

**FIG 3** A dual reporter strain highlights the effect of Z-FA-FMK on trypomastigogenesis. (A) Representative fluorescence images of the different *T. cruzi* life stages of the TdTom/ama-mNG expressing parasites showing specific mNG expression at the amastigote stage. Scale bar = 5 µm. (B) Representative fluorescence images of *T. cruzi*-infected HFFs at 4, 5, and 6 dpi showing the loss of mNG fluorescence as the parasites undergo trypomastigogenesis. Scale bar = 10 µm. (C) Representative fluorescence images of *T. cruzi*-infected HFFs at 6 dpi showing a (mNG+)-infected cell (yellow arrowhead) and a (mNG−)-infected cell (white arrowhead). Scale bar = 10 µm. (D) Graph showing the quantification of (mNG+)-infected cells at 5 and 6 dpi in culture treated or not with Z-FA-FMK. Statistical analysis was performed using multiple unpaired *t*-tests (mean ± SD, *n* = 3). (E) Quantification of the number of intracellular parasites per cell at 5 dpi, in culture treated or not with Z-FA-FMK (mean ± SD, *n* = 3; 75 cells/replicate). Statistical significance was assessed by the unpaired *t*-test.

abnormalities or differentiation arrest associated with Z-FA-FMK treatment, we performed a morphometric analysis on intracellular amastigotes. Measurement of average body length (excluding the flagellum), total flagellum length, and free flagellum length was obtained from U-ExM samples at 4 and 6 dpi and normalized to the internal expansion factor of each sample (Fig. 4C; Fig. S4A and B). Intracellular trypomastigotes at 6 dpi, from both treated and control conditions, were also analyzed using the same approach (Fig. S4C through G). Notably, the estimated unexpanded body length of amastigotes (4.8 ± 0.48 µm) and trypomastigotes (10.29 ± 0.50 µm) as well as the trypomastigote flagellum length (17.96 ±0.98 µm) were consistent with previous morphometric data obtained via conventional fluorescence microscopy and transmission electron microscopy (32, 33). However, the estimated flagellum length of amastigotes was slightly shorter (1.88 ± 0.20 µm) than previously reported values obtained by video-fluorescence microscopy (2.7 ± 0.70 µm) (34). Importantly, no significant differences were detected in the body length, total flagellum length, or free flagellum length between Z-FA-FMK-treated and control amastigotes at 4 or 6 dpi (Fig. 4D through F). Similarly, the few intracellular trypomastigotes present in Z-FA-FMK-treated cultures were morphologically indistinguishable from those in control conditions (Fig.

S4E through G). These findings demonstrate that the amastigotes accumulating under Z-FA-FMK treatment retain typical amastigote morphology, with no detectable structural abnormalities. Thus, Z-FA-FMK treatment leads to the accumulation of immature forms, being arrested at an early stage of trypomastigogenesis, supporting a role for cysteine proteases in the progression of amastigote to trypomastigote differentiation.

## DISCUSSION

The proliferation of *T. cruzi* within its mammalian host depends on iterative cycles of host cell invasion, intracellular multiplication, and release of newly formed parasites into the extracellular space. Throughout these stages, the parasite establishes close interactions with the host cell, potentially affecting host cell adhesion, morphology, and viability (4, 35–37). In this work, we used cellular impedance measurements to monitor, in real time, the effects of *T. cruzi* infection on host cells throughout the lytic cycle. Our results reveal a characteristic host cell response with three distinct phases, likely corresponding to invasion, multiplication, and late-stage processes of the lytic cycle. During the first 2.5 h of infection, we observed a sharp decrease in cellular impedance. This effect appears to extend ~30 min beyond the last invasion event, following the removal of extracellular parasites. By 5 hpi, impedance returned to control levels. This transient effect is likely linked to temporary actin depolymerization triggered by elevated cytosolic $Ca^{2+}$ released from intracellular stores, which facilitates *T. cruzi* invasion (5). The subsequent reassembly of F-actin has been shown to be required for intracellular retention of the parasite in non-phagocytic cells (3, 5). From 18 to ~80 hpi, infected cells consistently showed slightly higher impedance than uninfected controls. While our results do not pinpoint to a specific cause, previous studies suggest that *T. cruzi* infection alters gap junctions and induces a constant remodeling of the actin cytoskeleton to accommodate parasite expansion, both of which could directly impact cellular impedance (38, 39). Additionally, mitochondrial dysfunctions and infection-induced cellular stress may contribute indirectly to impedance changes during this period (40, 41). Starting at 4 dpi, impedance decreases steadily until reaching a plateau at 6 dpi. This triphasic impedance

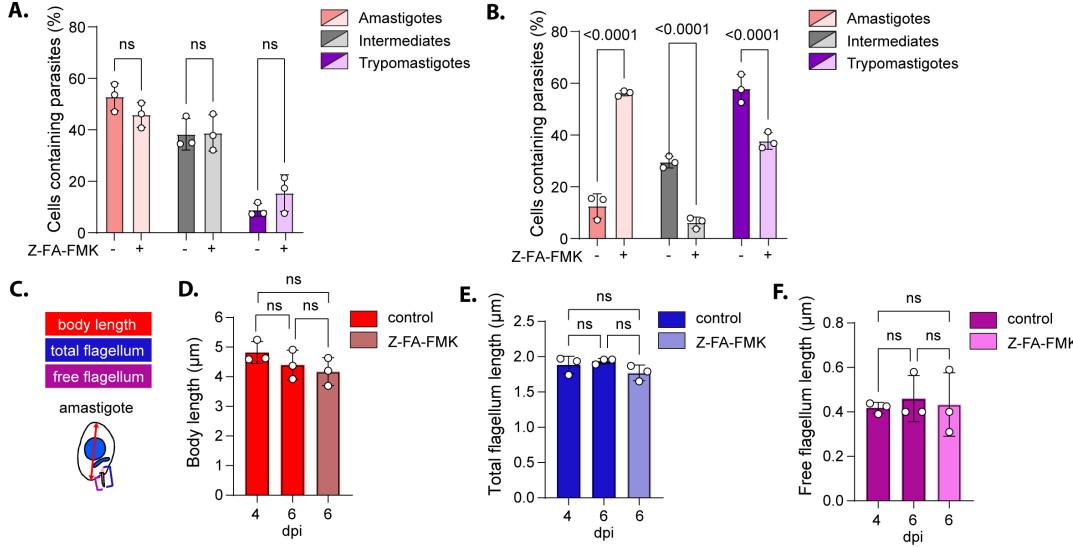

**FIG 4** Z-FA-FMK is an early-stage trypomastigogenesis inhibitor. (A) Graph representing the quantification of the different intracellular forms observed by U-ExM at 5 dpi in culture treated or not with Z-FA-FMK (mean ± SD, *n* = 3). Statistical significance was assessed by a two-way ANOVA followed by Šídák's multiple comparisons. (B) Graph representing the quantification of the different intracellular forms observed by U-ExM at 6 dpi in culture treated or not with Z-FA-FMK (mean ± SD, *n* = 3). Statistical significance was assessed by a two-way ANOVA followed by Šídák's multiple comparisons. (C) Schematic representation of the morphometric analysis performed on intracellular amastigotes from the U-ExM samples. (D–F) Graph representing the measurement of amastigote body length, total flagellum length, and free flagellum length at 6 dpi in culture treated or not with Z-FA-FMK (mean ± SD, *n* = 3). Statistical significance was assessed using ordinary one-way ANOVA with Tukey's multiple comparison (mean ± SD, *n* = 3). ns, not significant.

profile could be leveraged to evaluate mutant phenotypes and identify which step of the lytic cycle is affected, especially since six 96-well plates can be monitored simultaneously. To better resolve the last phase of the lytic cycle, we applied U-ExM to infected cells. Quantification of the different parasite forms shows that trypomastigogenesis is a prolonged process, beginning at 4 dpi and concluding at 6 dpi. The culmination of differentiation coincides with the release of mature trypomastigotes from the host cells at 6 dpi.

We next investigated the effect of Z-FA-FMK, a previously reported inhibitor of differentiation and egress (23, 26). Addition of Z-FA-FMK at 4 dpi impaired trypomastigogenesis, leading to an accumulation of infected cells containing amastigotes at 6 dpi. Morphometric analysis of U-ExM samples showed that these amastigotes retained the morphology of "classic" amastigotes, with no detectable signs of differentiation. This finding was reinforced by the prolonged expression of a reporter gene under the control of the amastin 3′UTR region in Z-FA-FMK-treated cultures. Amastin belongs to a family of surface glycoproteins specifically expressed at the amastigote stage (31). Inhibition of trypomastigogenesis delayed egress, with ~90% fewer parasites released at 6 dpi in treated cultures compared to control. Previous studies using cysteine protease inhibitors have consistently reported a reduction in the number of *T. cruzi* trypomastigotes released into the culture supernatant, leading to the interpretation that cysteine proteases, including cruzipain, may contribute to parasite egress from host cells (23, 26). In particular, Costales et al. (23), working with the *T. cruzi* Brazilian strain, reported a strong reduction in parasite egress following treatment with fluoromethyl ketone inhibitors added at day 4 post-infection, a time point at which differentiation into trypomastigotes was presumed to be complete based on morphological observations and neuraminidase staining (23). In contrast, our quantitative, time-resolved analyses reveal that, in the *T. cruzi* Y strain, trypomastigogenesis and egress are tightly coordinated and largely overlapping processes, such that inhibition of cysteine proteases during this late intracellular phase primarily impairs parasite maturation rather than host cell rupture, as reported by Hart et al. (26). The apparent discrepancy between these studies is likely explained by differences in parasite strain and differentiation kinetics. Applying the quantitative, time-resolved approaches described here to additional *T. cruzi* strains would be of particular interest to determine how variability in differentiation kinetics influences the temporal relationship between trypomastigogenesis and egress. We hypothesize that, in systems where differentiation precedes egress by a wider temporal margin, protease inhibition may predominantly directly affect parasite release, whereas in strains having a similar temporal relationship between trypomastigogenesis and egress to Y strain, inhibition would primarily block trypomastigogenesis. Together, these findings underscore the importance of quantitatively resolving differentiation and egress dynamics when interpreting pharmacological perturbations of the late *T. cruzi* lytic cycle.

Interestingly, the decrease in cellular impedance starting at 4 dpi was not fully blocked by Z-FA-FMK treatment, suggesting that host cell responses, possibly linked to actin cytoskeletal rearrangement described by Ferriera et al. (22) or host cell deterioration, occur partly independently of parasite differentiation and host cell rupture. Ultimately, Z-FA-FMK-treated cells released amastigotes continuously from 7 to 13 dpi. The increase in the number of extracellular parasites beyond 6 dpi likely reflects cumulative parasite release over time and may result from a combination of non-mutually exclusive mechanisms, including delayed rupture of initially infected host cells, reinvasion of freshly released parasites into susceptible host cells, potentially including cells already infected, leading to secondary rounds of host cell lysis, and/or completion of subsequent lytic cycles (most likely around 11–13 dpi).

Until now, egress efficiency has been assessed by counting parasites released in the culture supernatant using a hemocytometer or automated counter. Here, we developed an image-based phenotypic assay to quantitatively assess *T. cruzi* egress. This method relies on automated detection of extracellular parasites in the culture supernatant

using imaging and analysis software. While automated assays based on luminescence or DAPI staining have been applied to quantify parasite proliferation (42–46), to our knowledge, this is the first automated pipeline designed to measure egress efficiency in *T. cruzi*. This workflow provides a fast and efficient way to measure egress, while minimizing inter-operator variability. However, the unique cellular architecture of *T. cruzi* imposes inherent limitations on this method. The parasite contains two spatially distinct DNA-containing structures, the nucleus and the kinetoplast, which can occasionally be segmented as separate objects, leading to overestimation of parasite numbers. Segmentation parameters were empirically optimized to reduce this error without increasing the likelihood of erroneously merging closely adjacent parasites, particularly at higher densities. Although minor inaccuracies in absolute parasite counts may persist, the relative trends between experimental conditions were preserved and confirmed with manual counting, supporting the reliability of the automated method. Future refinements could include the implementation of advanced segmentation algorithms, such as object pairing rules, to further enhance accuracy.

Importantly, because the automated method does not require fixation of the infected monolayer, egress kinetics can be monitored dynamically by measuring parasite release at multiple time points. This workflow is versatile, does not require the generation of transgenic parasites, and can be applied to various strains as well as clinical isolates. Performed in a 384-well format, it is scalable for medium- to high-throughput screening of compounds or mutant libraries, supporting both drug discovery and target identification. Furthermore, when coupled with existing proliferation assays that use a similar pipeline, it enables a dual phenotypic screen capable of distinguishing compounds that inhibit proliferation versus egress (43).

In summary, our study establishes a scalable screening platform to identify key molecular players in the late stage of the *T. cruzi* lytic cycle. Complementary phenotypic assays allow discrimination between trypomastigogenesis and egress, while morphometric analyses provide precise identification of the specific differentiation steps affected. We leveraged this toolbox to re-examine in greater detail the role of Z-FA-FMK, demonstrating that it halts parasite differentiation at the early stage of trypomastigogenesis.

## MATERIALS AND METHODS

### Cell culture

Human foreskin fibroblasts (HFFs, ATCC SCRC-1041) and NRK-52E cells (ATCC, CRL-1571) were maintained in Dulbecco's modified Eagle's medium (DMEM-HPXA, Capricorn Scientific) supplemented with 10% heat-inactivated fetal bovine serum (FBS; Capricorn Scientific), 100 units/mL penicillin, 10 μg/mL streptomycin, and 2 mM glutamine (Capricorn Scientific). The cells were cultured under 5% $CO_2$ at 37°C.

### Parasite culture

Epimastigotes of *T. cruzi* Y strain (ATCC, 50832) were maintained in log-phase growth (<5 × $10^7$ parasites/mL), as determined by manual counting on a Neubauer counting chamber. Epimastigotes were cultured axenically at 27°C in liver infusion tryptose (LIT) medium supplemented with 10% heat-inactivated fetal bovine serum (FBS; Capricorn Scientific) (40). Mammalian infective stages were produced by prolonged starvation of epimastigotes at stationary phase until a large proportion of the culture contains metacyclic trypomastigotes (>80%). NRK-52E cells were infected with metacyclic trypomastigotes in DMEM supplemented with 2 mM glutamine, 100 units/mL penicillin, 10 μg/mL streptomycin, and 2% FBS (D2 medium). The infected culture was washed extensively every day to remove the remaining epimastigotes. Following the completion of the first lytic cycle (approximately 7 days post-infection with metacyclic), tissue culture-derived trypomastigotes (TCTs) were observed in the culture supernatant.

The mammalian stages of *T. cruzi* Y strain were maintained by weekly passage in NRK-52E cells cultured in DMEM supplemented with 2 mM glutamine, 100 units/mL penicillin, 10 µg/mL streptomycin, and 2% FBS (D2 medium) at 37°C, 5% $CO_2$ (10). Infective trypomastigotes were obtained from infected NRK-52E monolayers at the peak of egress. Supernatants were centrifuged at 250 × *g* for 5 min to remove host cells, followed by a centrifugation at 2,500 × *g* for 10 min to pellet parasites. The pellet was incubated for 3 h at 37°C to allow active trypomastigotes to swim upwards. The lower half of the supernatant, enriched in motile parasites, was collected for infection at a multiplicity of infection (MOI) of 20, if not stated otherwise, resulting in an infection rate of ~37%. At 2 h post-infection, the infected cultures were washed twice with PBS to remove extracellular parasites, and infected cells were maintained in D2 medium until use.

## Primers used in this study

1. tgctctataagttgtcttgtctagaATGGTTTCCAAAGGGGAAGAGGATAACA
2. tggctcgaggtcgacggtatcgataagcttTTATTTGTATAACTCATCCATTCCCATAACGT
3. cttatcgataccgtcgacctcgagCGGGTGCATCCACCGTCTGCATGC
4. ccttggagtcgtaaatggctcgagCGCAGGGCGGGCAGCGG

## Generation of TdTom/mNG parasites

To generate the plasmid pTREX-mNG-3'UTRama-bsd, the mNG coding sequence was PCR-amplified from mNG_3xHA_Bleo (generously provided by Dr. Lorenzo Brusini) using primers 1 and 2 and cloned by Gibson assembly into the plasmid TdTomato/pTrex-b (Addgene, 68709) previously digested by XbaI/HindIII. A fragment corresponding to the 803 bp of the 3' UTR of TcYC6_0087720 (*amastin*) was PCR-amplified, using primers 3 and 4, from genomic DNA of *T. cruzi* Y strain parasites and cloned into the mNG/pTrex-b at the XhoI restriction site by Gibson assembly. The resulting plasmid pTREX-mNG-3'UTRama-bsd was verified by sequencing.

For transfection, $4 \times 10^7$ Y strain epimastigotes in early log phase ($5 \times 10^6$ parasites/mL) were resuspended in 100 µL of Tb-BSF transfection buffer (90 mM Na-phosphate [pH 7.3], 5 mM KCl, 0.15 mM $CaCl_2$, 50 mM HEPES [pH 7.3]) (47) and mixed with 10 µg of pTREX-mNG-3'UTRama-bsd and pTrex-Neo-TdTomato (Addgene, 47975). Transfection was performed using the Amaxa Nucleofector II (program X-014) and transfected parasites were cultured in LIT-10% FBS. Transgenic parasites were selected by the addition of neomycin (250 µg/mL) and blasticidin (10 µg/mL), 1 day post-transfection. Growth was monitored to maintain the parasite in log phase. One month post-transfection, fluorescence in the transfected culture was assessed by flow cytometry, revealing over 70% positive cells. Parasites were then serially cloned twice into 96-well plates by limiting dilution under drug selection pressure. Positive clones were selected based on integration PCR results and fluorescence analysis.

## Quantification of (mNG+)-infected cells

Isolated *T. cruzi* trypomastigotes were used to infect confluent HFF on glass coverslip at an MOI of 10, resulting in an infection rate of ~15.7% (see Parasite culture). At 4 dpi, the culture medium was replaced by D2 containing either 1% DMSO or Z-FA-FMK (100 µM) in 1% DMSO. Coverslips were fixed at 4, 5, and 6 dpi with 4% paraformaldehyde (PFA, Electron Microscopy Sciences), stained with 4',6-diamidino-2-phenylindole (DAPI, Invitrogen), and analyzed on the confocal microscope DM5500B (Leica) at the Microscopy Imaging Center of the University of Bern. Infected host cells were classified as mNG+ if they contained 50% or more mNG+ parasites. Infected cells containing less than 50% of mNG+ parasites were classified as mNG− cells. For each condition, at least 100 infected cells were counted. Experiments were performed in three independent biological replicates. Results are presented as mean ± SD. Statistical analysis was performed using multiple unpaired *t*-tests on GraphPad Prism 8 software.

## Multiplication assay

HFF monolayers were seeded on glass coverslips in 24-well plates and infected with *T. cruzi* tdTomato/ama-mNeonGreen trypomastigotes at a multiplicity of infection (MOI) of 10, resulting in an infection rate of ~15.7% (see Parasite culture). At 4 dpi, cultures were treated with either 50 µM Z-FA-FMK (ApexBio) or vehicle control (0.1% DMSO in D2 medium). After 24 h of treatment (5 dpi), coverslips were fixed in 4% paraformaldehyde (Electron Microscopy Sciences) and stained with DAPI (Invitrogen).

Images were acquired with a Leica DM5500B fluorescence microscope at the Microscopy Imaging Center of the University of Bern. Image analysis was performed in Fiji (ImageJ 1.54f, NIH, Bethesda, USA). The DAPI channel was thresholded, binarized, and segmented using the watershed algorithm to separate individual parasites. Parasite nuclei were identified and counted with the "Analyze Particles" function using a size filter appropriate for *T. cruzi* amastigotes. Host cell nuclei were used only to assign parasites to individual cells. For each replicate and condition, intracellular parasite burden was determined by quantifying amastigotes in 75 host cells. Results are expressed as the mean number of parasites per cell. Three technical replicates were performed. Statistical analysis was carried out in GraphPad Prism v10 (GraphPad Software) using the Mann–Whitney test.

## Kinetic assay measuring host cell impedance

Host cell impedance during the infection was assessed by the Live Cell Analysis System XCelligence (OLS) and analyzed on the RTCA software (Agilent, version 1.0). HFFs were plated in E-8 well plate at a density of $5 \times 10^4$ cells/well and cultured until confluency in D10 medium. Cellular impedance was measured every 15 min during the entire experiment. On the day of infection, D10 medium was replaced by D2 medium for at least 3 h. Wells were infected with $1 \times 10^6$ parasites, and the plate was replaced in the xCELLigence for measurement. Two hours post-infection, remaining extracellular parasites were removed by three washes in D2 medium. At 4 dpi, medium was replaced by medium containing 0.25% DMSO or 50 µM Z-FA-FMK (ApexBio), and cellular impedance was measured every 15 min during three additional days. For the analysis of toxicity, benznidazole (Merck) and Z-FA-FMK (ApexBio) were used at different concentrations. Data are normalized to the cellular impedance at the time of infection and expressed as a percentage of the impedance measured for control wells over time. For the time course of infection and the effect of Z-FA-FMK on egress, the experiments were done in four independent biological replicates. For the toxicity tests, two independent biological replicates were done.

## Manual and automated quantification of egress

### Infection

HFF monolayers in 384-well plates (Corning, 3764) were infected with *T. cruzi* trypomastigotes in D2 medium for 2 h at 37°C and 5% $CO_2$. Non-internalized parasites were removed by two washes with Dulbecco's PBS (Capricorn Scientific), and cultures were maintained in D2 without phenol red (Capricorn Scientific, DMEM-HXRXA). For drug assays, cultures at 4 dpi were treated with Z-FA-FMK (50 µM; ApexBio, B1360), benznidazole (3.125 µM; Merck, 419656), or DMSO (0.1%) as a vehicle control. Untreated infections were maintained under the same conditions without compound addition.

### Sampling and imaging

For untreated infections, half of the culture supernatants was transferred to µClear 384-well plates (Greiner Bio-One, 781986) at 3, 4, 5, 6, and 7 dpi and fixed by adding paraformaldehyde (32% aqueous; Electron Microscopy Sciences, 15714) in PBS to a final concentration of 4%, together with Hoechst 33342 (ThermoFisher Scientific, 62249) at a 1:5,000 dilution in a total volume of 80 µL per well. For drug-treated cultures, culture

supernatants were collected at 4, 6, 7, 9, 11, and 13 dpi and processed similarly. At each time point, representative images of both the monolayer (MN) and supernatant (SN) were acquired with an Olympus IX83 inverted fluorescence microscope equipped with a CoolLED pE-300white illumination system (CoolLED Ltd., Andover, UK).

## Quantification of extracellular parasites

For manual counts, 10 µL of fixed SN was loaded into a Neubauer chamber, and parasite numbers per well were calculated as counted parasites ×800, where 800 is the multiplication factor accounting for chamber volume and dilution. For automated counts, fixed, Hoechst-stained SN plates were imaged with the In Cell Analyzer 2000 (Cytiva) with 12 images per well (covering ~85% of the well area) at an exposure of 0.100 s with the DAPI channel, and using a 20× objective. Hoechst-positive objects were quantified using IN Cell Developer Toolbox software (Cytiva, version 1.10.0). *Trypanosoma cruzi* contains two spatially separated DNA-containing structures, the nucleus and the kinetoplast, both labeled by Hoechst. During automated image analysis, these structures can occasionally be segmented as independent objects, resulting in overestimation of parasite numbers. To mitigate this, we implemented a distance-based merging step in which Hoechst-positive objects within a defined spatial threshold were counted as a single parasite. The distance threshold was empirically optimized based on manual measurements of nucleus–kinetoplast spacing. Increasing this threshold reduced over-segmentation of individual parasites but led to occasional merging of closely adjacent independent parasites, an effect that increased with parasite density. Therefore, a conservative threshold was selected to minimize erroneous merging while acknowledging that some parasites may still be counted as separate objects. The parameters used in this study were as follows: Kernel size = 7, sensitivity = 30, and a size filter of $3\ \mu m^2 < X < 30\ \mu m^2$) was applied to exclude Hoechst-positive host cell nuclei or debris.

## Data processing and statistical analysis

For both experimental approaches, three independent biological replicates were performed, each containing three technical replicates. For manual counting, the proportion of each developmental form (amastigotes, trypomastigotes, and intermediate forms) relative to the total parasite count was also determined using the general morphology observed by transmitted light. For automated counting in untreated cultures, raw extracellular parasite numbers are reported. In drug-treated cultures, extracellular parasite counts were normalized to the mean value of the corresponding DMSO control for each time point. Statistical analyses were performed in GraphPad Prism v10 (GraphPad Software) using multiple *t*-tests, one-way ANOVA, or two-way ANOVA, as indicated in figure legends.

## Ultrastructure-expansion microscopy

Coverslip of HFF monolayer infected with *T. cruzi* was fixed in 1.4% formaldehyde/2% acrylamide/PBS solution for 5 h at 37°C. The gelation step was performed on ice with the gelation solution (90 µL monomer solution (19% sodium acrylate [Sigma]/10% acrylamide [Sigma]/0.1% N,N′-methylenbisacrylamide [Sigma]/PBS) + 5 µL TEMED 10% + 5 µL APS 10%) in a humid chamber for 5 min followed by an incubation at 37°C for 1 h. The embedded coverslip was then incubated in denaturation buffer (200 mM SDS, 200 mM NaCl, and 50 mM Tris pH 9.0) for 15 min at room temperature under agitation until the gel detaches from the coverslip. Following a denaturation step performed at 95°C for 1.5 h, the gel was subjected to a first round of expansion in ddH$_2$O for 30 min (repeat twice) followed by two rounds of gel shrinkage in PBS for 15 min. For immuno-detection of microtubules, the gels were incubated for 3 h at 37°C with anti-alpha and beta tubulin diluted 1/250 in BSA 2%/Tween 0.1%/PBS (kind gift from Prof. Soldati-Favre) (48). Following three washes in Tween 0.1%/PBS and secondary antibody detection (Invitrogen), a last round of expansion was done by incubating the gel overnight in

ddH$_2$O. For imaging, pieces of gel were put on poly-D-lysine (Gibco) coated 24 mm coverslip with sample facing down clipped on 35 mm round adapters (Okolab). Images were acquired on the confocal microscope TCS SP8 microscope (Leica) with the 63× 1.4 NA oil immersion objective at the Microscopy Imaging Center of the University of Bern. Images were processed using ImageJ software (NIH; version 1.53c). For the different stages, 100 infected cells were counted per sample, in three independent biological replicates. Results are presented as mean ± SD. Statistical analysis was performed using two-way ANOVA followed by Šídák's multiple comparisons test on GraphPad Prism 8 software.

## Morphometric analysis

### Data processing and measurement

Confocal image stacks were processed using ImageJ (Fiji distribution, version 1.54p, NIH) with custom macros (Lentini G. 2025. 1_morphometric analysis_final and 2_morphometric analysis_final. GitHub https://github.com/gaellelentini/Morphometric-analysis). In the first macro, fluorescence channels were split, and each channel was subjected to a maximum intensity Z-projection. Channel 1, corresponding to tubulin, was converted to magenta, while Channel 2, corresponding to DNA, was converted to grayscale. Both channels were converted to 8-bit and contrast was enhanced, and the channels were then merged. For manual annotation, the composite images were imported into Procreate (version 5.3.15, Savage Interactive Pty Ltd). On three independent transparent layers, the following parasite features were traced: body length, total flagellum length, and free flagellum length. The annotated layers were exported separately containing only the traced lines used for analysis. Each annotation file was re-imported into ImageJ and processed with the second macro. Images were converted to 8-bit, contrast-enhanced, and thresholded to generate binary masks. Masks were skeletonized, and scale calibration was set according to the microscope parameters (1 μm = 2.7693 pixels). Morphometric parameters were extracted using the Analyze Skeleton (2D/3D) plugin, which provided quantitative length measurements for each traced structure. For each condition, 30 individual parasites were measured.

### Extrapolated unexpanded size derived from expanded sample data

All morphometric measurements were obtained from expanded parasite samples. To recover the real biological dimensions, expansion factors were calculated based on nuclear size measurements. For unexpanded control parasites (DMSO-treated), 50 nuclei of uninfected HFFs were measured per time point (4, 5, and 6 days post-infection), resulting in a total of 150 nuclei. The mean value served as the reference for true nuclear dimension. In the expanded samples, 20 nuclei were measured for each condition (4 and 6 dpi, with or without drug treatment). Expansion factors were calculated independently for each day and treatment condition by comparing the mean nuclear size in the corresponding expanded samples with the unexpanded control value. These condition-specific factors were then applied to all morphometric features (body length, total flagellum length, and free flagellum length) to obtain corrected, real-size measurements of the parasites.

## ACKNOWLEDGMENTS

We thank Dr. Reto Caldelari (University of Bern) for his help regarding the development of the image-based assay. We thank Prof. Dr. Jürg Gertsch and Prof. Dr. Olivier Pertz for sharing the *T. cruzi* TC6 strain and NRK-52E cells (University of Bern). We would like to thank Prof. Dr. Soldati-Favre for sharing human foreskin fibroblast and the anti-alpha/beta tubulin hybridoma (University of Geneva, Geneva Antibody Facility). We thank Dr. Brusini for providing us with the mNG_3xHA_Bleo plasmid (University of Geneva) and M. Yves Cambet for the loan of the xCELLigence device (READS unit, University of Geneva).

## AUTHOR AFFILIATIONS

[1]Institute of Cell Biology, University of Bern, Bern, Switzerland

[2]Graduate School for Cellular and Biomedical Sciences, University of Bern, Bern, Switzerland

## AUTHOR ORCIDs

Gaelle Lentini ⬤ http://orcid.org/0000-0003-0836-9972

## FUNDING

| Funder | Grant(s) | Author(s) |
| --- | --- | --- |
| Schweizerischer Nationalfonds zur Förderung der Wissenschaftlichen Forschung | PZ00P3_209016 | Gaelle Lentini |

## AUTHOR CONTRIBUTIONS

Sara De Grandis, Data curation, Formal analysis, Investigation, Methodology, Validation, Visualization, Writing – review and editing | Anne Niggli, Data curation, Formal analysis, Investigation | Delia Bogenstätter, Data curation, Formal analysis, Software, Writing – review and editing | Gaelle Lentini, Conceptualization, Data curation, Formal analysis, Funding acquisition, Investigation, Methodology, Project administration, Resources, Supervision, Validation, Visualization, Writing – original draft, Writing – review and editing

## ADDITIONAL FILES

The following material is available online.

### Supplemental Material

**Supplemental material (Spectrum04132-25-s0001.pdf).** Fig. S1 to S4.

### Open Peer Review

**PEER REVIEW HISTORY (review-history.pdf).** An accounting of the reviewer comments and feedback.

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
