## [Reviewer comments · Microbiology Spectrum]

Microbiology Spectrum

Revisiting Cysteine Protease Function in *Trypanosoma cruzi*: Implications for Parasite Egress and Differentiation

Sara De Grandis, Anne Niggli, Delia Bogenstätter, and Gaëlle Lentini

Corresponding Author(s): Gaëlle Lentini, Universitat Bern

Review Timeline:

Submission Date:	January 14, 2026
Editorial Decision:	March 3, 2026
Revision Received:	March 4, 2026
Accepted:	March 26, 2026

Editor: Galadriel Hovel-Miner

Reviewer(s): The reviewers have opted to remain anonymous.

Transaction Report:

DOI: <https://doi.org/10.1128/spectrum.04132-25>

Re: Spectrum04132-25 (**Revisiting Cysteine Protease Function in *Trypanosoma cruzi*: Implications for Parasite Egress and Differentiation**)

Dear Dr. Gaëlle Lentini:

Thank you for the privilege of reviewing your work. Below you will find my comments, instructions from the Spectrum editorial office, and the reviewer comments.

I am pleased to inform you that your manuscript has been editorially accepted for publication. However, there are a few additional questions in the submission form that need to be answered before the final decision. Once these are completed, please return your submission so that I can move your paper forward to acceptance.

Sincerely,
Galadriel Hovel-Miner
Editor
Microbiology Spectrum

Reviewer #1 (Comments for the Author):

The manuscript entitled: "Revisiting Cysteine Protease Function in *Trypanosoma cruzi*: Implications for Parasite Egress and Differentiation" by De Grandis and collaborators described a new method for the analysis and quantification of trypomastigogenesis in the lytic cycle of *Trypanosoma cruzi*, the etiologic agent of Chagas disease. Trypomastigogenesis is developmental transformation from replicative amastigotes to non-dividing, infective trypomastigotes, a process that the parasite completes intracellularly before egressing from the mammalian host cell. This process is poorly understood in *T. cruzi*. While quantifying host cell invasion by cell-derived trypomastigotes and intracellular replication of amastigotes has been performed by many laboratories using well established methods involving cell fixation and image acquisition, the quantification of trypomastigogenesis is mainly performed through the quantification of the number of trypomastigotes present in the supernatant of infected cells between days 4 and 8 post infection. This methodology does not allow to discriminate the trypomastigogenesis from the egress phenotype of a specific *T. cruzi* cell line. In the present study the authors developed a methodology to quantify trypomastigogenesis using a combination of cell impedance monitoring, stage-specific fluorescent parasites, and automated high-content imaging. The methods were elegantly developed, and the statistical analysis of the results is robust. The authors used their method to evaluate the effect of a potent cysteine protease inhibitor (Z-FA-FMK) on trypomastigogenesis. In general, the results are very well presented, and the methods were properly standardized. The contribution of this study is undoubtedly valuable to the *T. cruzi* community. I do not have further critiques on the present version of the manuscript.

Reviewer #2 (Comments for the Author):

The authors have successfully addressed all points raised in my previous review. I particularly appreciate the thoughtful revision of the manuscript to better contextualize the findings within the existing literature, specifically regarding the coordination of differentiation and egress. The added discussion clarifying the strain specific kinetics of Z-FA-FMK provides excellent clarity on the mechanistic differences compared to prior reports. This study establishes a rigorous, quantitative framework that will be a valuable contribution to the *T. cruzi* community, and I have no further comments.

Reviewer #3 (Comments for the Author):

The authors have adequately addressed this reviewer's comments from a previous review and are congratulated for a thorough

study that brings new tools to the field and addresses limitations of earlier studies.

Re: Spectrum04132-25R1 (**Revisiting Cysteine Protease Function in *Trypanosoma cruzi*: Implications for Parasite Egress and Differentiation**)

Dear Dr. Gaëlle Lentini:

Your manuscript has been accepted, and I am forwarding it to the ASM production staff for publication. Your paper will first be checked to make sure all elements meet the technical requirements. ASM staff will contact you if anything needs to be revised before copyediting and production can begin. Otherwise, you will be notified when your proofs are ready to be viewed.

Sincerely,
Galadriel Hovel-Miner
Editor
Microbiology Spectrum

Reviewer #2 (Comments for the Author):

All comments and concerns have been addressed.

Reviewer #3 (Comments for the Author):

The authors have adequately addressed my comments and are congratulated for an excellent study.